# Causal Discovery in Heterogeneous Environments Under the Sparse Mechanism Shift Hypothesis

**Ronan Perry**[1], **Julius von Kügelgen**[* 1,2], **and Bernhard Schölkopf** [*1]

[1]Max Planck Institute for Intelligent Systems, Tübingen, Germany
[2]University of Cambridge, United Kingdom
`rflperry@uw.edu, jvk@tue.mpg.de, bs@tue.mpg.de`

## Abstract

Machine learning approaches commonly rely on the assumption of independent and identically distributed (*i.i.d.*) data. In reality, however, this assumption is almost always violated due to distribution shifts between environments. Although valuable learning signals can be provided by heterogeneous data from changing distributions, it is also known that learning under arbitrary (adversarial) changes is impossible. Causality provides a useful framework for modeling distribution shifts, since causal models encode both observational and interventional distributions. In this work, we explore the *sparse mechanism shift hypothesis*, which posits that distribution shifts occur due to a *small* number of changing causal conditionals. Motivated by this idea, we apply it to learning causal structure from heterogeneous environments, where i.i.d. data only allows for learning an equivalence class of graphs without restrictive assumptions. We propose the *Mechanism Shift Score* (MSS), a score-based approach amenable to various empirical estimators, which provably identifies the entire causal structure with high probability if the sparse mechanism shift hypothesis holds. Empirically, we verify behavior predicted by the theory and compare multiple estimators and score functions to identify the best approaches in practice. Compared to other methods, we show how MSS bridges a gap by both being nonparametric as well as explicitly leveraging sparse changes.

## 1 Introduction

Classical machine learning methods and theory presume data to be independently and identically distributed (*i.i.d.*). Although there has been huge success under this assumption, research on topics including adversarial examples [17, 58], distribution shifts [9, 46, 48, 50], and "spurious" correlations [2] has highlighted its fragility. Open questions remain as to how we can relax the i.i.d. assumption and still learn useful models, since learning under unrestricted adversarial distribution shifts seems infeasible [7]. Causal models naturally provide structure to a distribution via a factorization into causal *mechanisms*, the processes by which variables are dependent on their direct causes. Hence, they are a natural basis for studying distribution shifts. Based on the idea of the independence of causal mechanisms [41, 50], the *sparse mechanism shift hypothesis* [51] posits that distribution shifts are the result of changes in only a subset of the causal model's mechanisms. This presents a promising relaxation with many potential applications throughout machine learning [3, 36, 37, 51, 61].

In particular, causal discovery—the inference of qualitative structure encoded by a graph and underlying causal relationships between variables—is a fundamental scientific problem with broad applications for which distribution shifts have been successfully leveraged. Classical approaches, which assume i.i.d. data from a single environment or domain, can be broadly categorized into three

---

[*]Shared last author.

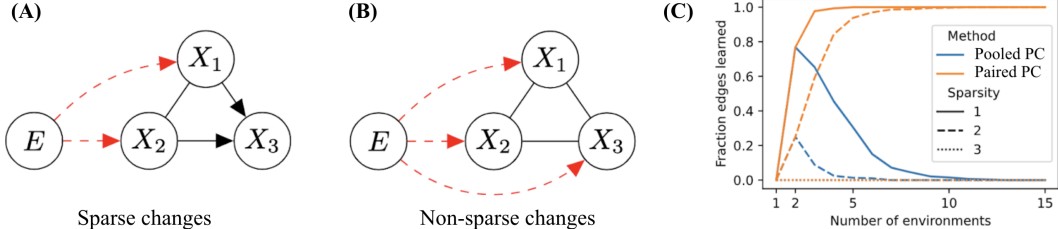

Figure 1: **Sparse shifts yield structure identifiability**. In a causal model with a fully connected DAG over $\mathbf{X} := \{X_1, X_2, X_3\}$, no edge directions can be learned from i.i.d. observational data. Given multiple datasets on $\mathbf{X}$ across different environments with possible distribution shifts, the PC algorithm [56] can be applied to the pooled data augmented by the environment index variable $E$ [28]. **(A)** A mechanism shift $E \rightarrow X_i$ can allow orientation of some edges. **(B)** Dense shifts prohibit any orientations [28]. **(C)** Combining results learned across *pairs* of pooled environments leads to identifiability under *sparse* shifts. Pooling all environments leads to dense shifts, even if pairs of environments differ sparsely; under *dense* shifts, only the equivalence class is learned.

classes. *Constraint-based* methods, such as the PC [56] and FCI [42] algorithms, perform a series of statistical independence tests to identify the existence of causal relationships under various assumptions (e.g., causal faithfulness) and then use certain rules to infer causal directions. *Score-based* methods, such as GES [5], optimize some score function, such as the penalized likelihood BIC [52], over the set of possible graphs. These methods all come with asymptotic *structure identifiability* guarantees of recovering the (Markov) equivalence class [15], but they will rarely uniquely recover the true causal graph. Further work on *functional-form-based* methods, beginning with non-Gaussian assumptions [55], provides various ways to recover a specific DAG by assuming a certain functional form of the causal mechanisms [26, 30, 43, 63], at the cost of potential misspecification.

Although identification is possible through actively specified interventions [10, 11, 20, 21, 54], natural distribution shifts on the same variables across different environments allow for promising approaches under a relaxation of the i.i.d. assumption [8, 13, 14, 16, 18, 22, 23, 27, 28, 33–35, 38, 44, 50]. Such *multi-environment* methods can learn the direct causes of a specific variable [23, 44] or the entire causal structure [13, 14, 28] without requiring knowledge of which causal mechanisms change. Yet, as noted in these work but not studied, performance is dependent on which and how many changes occur.

**Overview and contributions.** After a review of causal graphical models and formalization of our setting (§ 2), we discuss key related work on multi-environment causal discovery (§ 3). We demonstrate that sparse distribution changes provide structure identifiability in the bivariate case, and useful information in multivariate settings (§ 4). Based on the observation that pairwise comparisons of environments better leverage sparse changes (see Fig. 1), we propose the *Mechanism Shift Score*, a score-based causal discovery algorithm with theoretical guarantees (§ 5). Empirical results confirm the theory (§ 6). In summary, the present work makes the following contributions:

- We prove that by relaxing the i.i.d. assumption via the sparse mechanism shift assumption, bivariate causal structure is identifiable without parametric assumptions (Cor. 4.2).
- We introduce the *Mechanism Shift Score (*MSS*)*, defined as the number of conditional distributions implied by a graph which change across all pairs of environments (§ 5). We prove that the true causal (multivariate) graph minimizes the MSS over possible graphs (Prop. 5.1).
- We provide rates of convergence showing that with a sufficient number of sparsely changing environments, the causal graph *uniquely* minimizes the score function with high probability (Cor. 5.4). Our rates readily apply to existing literature on learning individual mechanisms [23, 44] and the entire graph [14, 28], where a study of the role of sparsity was previously missing.
- We demonstrate empirically that sparsity and pairwise comparisons are useful and show how the MSS accommodates various parametric [13, 14, 44] and nonparametric [23, 28] estimators (§ 6).

## 2 Problem setting and notation

We start by building up the causal framework needed to understand our work and related literature. It relies on common graph-theoretic terminology which we review for completeness in Appx. A.

**Causal terminology.** Causal relationships between variables are encoded in a causal graphical model (CGM) which links graphical and distributional properties via certain assumptions.

**Definition 2.1** (Causal Graphical Model (CGM)). A causal graphical model (CGM) $\mathcal{M} = (G, \mathbb{P}_{\mathbf{X}})$ over $d$ random variables $\mathbf{X} = \{X_1, ..., X_d\}$ consists of (i) a *directed acyclic graph* (DAG) $G$ with vertices $\mathbf{X}$ and edges $X_i \to X_j$ iff $X_i$ is a direct cause of $X_j$, (ii) and a joint distribution $\mathbb{P}_{\mathbf{X}}$ which factorizes (is *Markovian*) over $G$.[2] Formally, we have the following *Markov* or *causal factorization*:

$$\mathbb{P}_{\mathbf{X}}(X_1, ..., X_d) = \prod_{j=1}^{d} \mathbb{P}_{\mathbf{X}}(X_j \mid \mathbf{PA}_j), \tag{2.1}$$

where $\mathbf{PA}_j$ are the parents (direct causes) of $X_j$ in $G$ and $\mathbb{P}_{\mathbf{X}}(X_j|\mathbf{PA}_j)$ is the *causal mechanism* of $X_j$.

Implicit in the definition is the assumption that there is *no hidden confounding*, i.e., that any common cause of two or more observed variables is included in $\mathbf{X}$; for this reason it is also referred to as *causal sufficiency*. This is a strong assumption to make and important to keep in mind in applications.

The Markov factorization (2.1) of the CGM encodes various conditional independences between variables. The *Markov equivalence class (MEC)* is the set of DAGs which share the same set of conditional independence relations; graphically, it is the set of DAGs which share the same *skeleton* (set of edges regardless of direction) and *v-structures* ($X_i \to X_j \leftarrow X_k$, but $X_i \not\leftrightarrow X_k$) [45, Lem. 6.25]. A set of DAGs such as the MEC are commonly represented as a *completed partially directed acyclic graph (CPDAG)* in which edges are directed if directed in all DAGs in the set, and otherwise left undirected [19]. Incorrect DAGs in the MEC induce *non-causal factorizations* containing *non-causal conditionals* which differ from the true causal mechanisms. Furthermore, in a CGM, equation (2.1) is equivalent to the *global Markov condition* which states that for disjoint vertex sets $\mathbf{A}, \mathbf{B}, \mathbf{Z}$ in $G$:

$$\mathbf{A} \perp\!\!\!\perp_G \mathbf{B} \mid \mathbf{Z} \implies \mathbf{A} \perp\!\!\!\perp \mathbf{B} \mid \mathbf{Z} \tag{2.2}$$

where $\perp\!\!\!\perp_G$ denotes d-separation in $G$ (Appx. A) [45, Thm. 6.22]. While the Markov property allows us to derive distributional properties from the DAG, causal discovery concerns deriving graphical properties from distributional properties. This requires the *causal faithfulness assumption*, effectively assuming that variables are not statistically independent unless implied by the graph.

**Assumption 2.2** (Causal faithfulness). The observational distribution $\mathbb{P}_{\mathbf{X}}$ is said to be *faithful* to the causal graph $G$ if every conditional independence relationship in $\mathbb{P}_{\mathbf{X}}$ implies d-separation in $G$ (i.e., d-connection implies statistical dependence [45, Def. 6.33]).

**Multi-environment data.** We assume that we observe a collection $\mathcal{D}$ of datasets from a set $\mathcal{E}$ of (possibly different) environments, where each dataset $\mathcal{D}^e$ from environment $e$ contains $n_e$ independent and identically distributed (i.i.d.) observations from some joint distribution $\mathbb{P}_{\mathbf{X}}^e$,

$$\mathcal{D} = \{\mathcal{D}^1, ..., \mathcal{D}^{n_{\mathcal{E}}}\} \quad \text{where} \quad \mathcal{D}^e = \{\mathbf{X}^{e,1}, ..., \mathbf{X}^{e,n_e}\} \overset{\text{i.i.d.}}{\sim} \mathbb{P}_{\mathbf{X}}^e \quad \text{for} \quad e \in \mathcal{E} = \{1, ..., n_{\mathcal{E}}\}.$$

Environments can encapsulate experimental settings, or broader contexts such as climate or time [38]. A key assumption of our setting is that environments arise from different "soft" interventions on an underlying shared CGM $\mathcal{M}$. Specifically, the CGM $\mathcal{M}$ entails a set of *interventional distributions*, resulting from changing a subset of the *mechanisms* $\mathbb{P}_{\mathbf{X}}(X_j|\mathbf{PA}_j)$ to some different $\tilde{\mathbb{P}}_{\mathbf{X}}(X_j|\mathbf{PA}_j)$.[3]

**Assumption 2.3** (Shared mechanisms). Each environment $e$ independently results from $\mathcal{M}$ by intervening on an (unknown) subset $\mathcal{I}^e \subseteq [d]$ of mechanisms, i.e., $\tilde{\mathbb{P}}_{\mathbf{X}}^e$ can be written as

$$\tilde{\mathbb{P}}_{\mathbf{X}}^e(X_1, ..., X_d) = \left(\prod_{j \in \mathcal{I}^e} \tilde{\mathbb{P}}_{\mathbf{X}}^e(X_j \mid \mathbf{PA}_j)\right) \prod_{j \in [d] \setminus \mathcal{I}^e} \mathbb{P}_{\mathbf{X}}(X_j \mid \mathbf{PA}_j). \tag{2.3}$$

We make the following common assumption about how these changes arise.

**Assumption 2.4** (Independent causal mechanisms (ICM) [41, 50]). A change in $\mathbb{P}(X_j \mid \mathbf{PA}_j)$ has no effect on and provides no information on $\mathbb{P}(X_k \mid \mathbf{PA}_k)$ for any $k \neq j$.

Since causal mechanisms are fixed within an environment and potentially vary across them, we can create an *augmented CGM* to unify CGMs from different environments. Note that in Asm. 2.3 and the following definition, the same causal parents and DAG are preserved over different environments; the distribution changes are limited to soft interventions which do not change a variable's causal parents.

---

[2]Throughout, we assume the existence of densities with respect to the Lebesgue measure.

[3]Note that this includes "hard" interventions $\tilde{\mathbb{P}}_{\mathbf{X}}(X_j \mid \mathbf{PA}_j) = \delta(X_j - x)$ which fix $X_j$ to a specific value $x$, and also "soft" interventions which merely alter the functional relationship.

**Definition 2.5** (Augmented CGM [28]). Let $\{(G, \mathbb{P}_{\mathbf{X}}^e)\}_{e=1}^{n_{\mathcal{E}}}$ be a collection of CGMs over the DAG $G = (\mathbf{X}, T)$ from multiple environments. The augmented CGM is defined as $\mathcal{M}' := (G', \mathbb{P}_{\mathbf{X} \cup E})$ where (i) $E$ is a random environment indicator variable with support $\mathcal{E}$, and (ii) the augmented DAG $G'$ has vertices $\mathbf{X} \cup E$ and edge set $T \cup \{(E, X_j) : \exists e, e' \in \mathcal{E} \text{ s.t. } \mathbb{P}_{\mathbf{X}}^e(X_j | \mathbf{PA}_j^G) \neq \mathbb{P}_{\mathbf{X}}^{e'}(X_j | \mathbf{PA}_j^G)\}$.

Note that $\mathbb{P}_{\mathbf{X} \cup E}$ is Markovian to $G'$, inheriting the factorization from the underlying CGMs along with the added dependence on $E$. With respect to the augmented DAG, Asm. 2.4 implies that existence of the edge $E \to X_i$ provides no information on the existence of an edge $E \to X_j$ for $j \neq i$. As discussed by Huang et al. [28], since $E$ can be a common cause of variables in $\mathbf{X}$, causal sufficiency in the original CGM over $\mathbf{X}$ is violated. We instead must assume *pseudo-causal sufficiency*. See Appx. C for further discussion of this assumption and the ICM principle.

**Assumption 2.6** (Pseudo causal sufficiency [28]). Any unobserved confounders of variables in $\mathbf{X}$ can be written solely as functions of $E$. Thus, within any given environment $e$, all unobserved confounders are fixed and causal sufficiency holds.

**Sparse mechanism shift (SMS) hypothesis.** Another key assumption, supported by Asm. 2.4's implication that a change to one mechanism does not imply changes to others, is the following:

**Assumption 2.7** (SMS [51]). Changes in mechanisms between observed environments are sparse:

$$0 < |\mathcal{I}^e| < d \tag{2.4}$$

The value of this assumption when met is illustrated in Fig. 1 and will be elaborated upon in § 5.

## 3 Related work on causal discovery from multiple environments

Causal discovery from changing distributions and causal mechanisms has a long history, going back to Simon's *invariance criterion*, stating that the true causal order is the one that is invariant under the right sort of intervention [24, 25]. Tian and Pearl [59] infer a causal order by testing which marginal distributions change under a single intervention. Invariant causal prediction (ICP) [44] can identify the causal parents of a target variable under the assumption that the target's causal mechanism is invariant across environments [23, 44, 50]. However, applying ICP to each variable in order to learn all sets of causal parents and hence the causal graph is not immediately admissible: the invariance assumption would imply that all variables are invariant, and thus there are no mechanism changes to learn from.

Learning the MEC from i.i.d. data is a well-studied problem, and under certain assumptions, essentially solved [5, 42, 56]. In our multi-environment setting, Yang et al. [62] characterized the $\mathcal{I}$-MEC, a subset of the MEC, which was subsquently generalized to a $\Psi$-MEC [29] under an unknown observational environment. Brouillard et al. [4] developed an estimator under unknown intervention targets, but none of these works implement an algorithm or perform experiments on soft interventions with unknown targets. Ghassami et al. [13] apply ideas from linear ICP to identify the causal DAG. Assuming *linear* causal mechanisms in which only the noise distributions change, they compare pairs of environments and orient edges to meet this requirement. Ghassami et al. [14] allow for any kind of change within a *linear* model, demonstrating that the causal DAG minimizes the number of changes in the linear model parameters. Squires et al. [57], like Jaber et al. [29], extend this approach to a nonparametric setting by applying existing methods to an augmented graph containing additional nodes for each pair of environments (see Appx. C.3 for a more detailed discussion). In contrast, Huang et al. [28] apply the PC algorithm on the augmented CGM, pooling all the data as in Mooij et al. [38]; this identifies the MEC but more importantly can also orient additional edges. Since not all edges are guaranteed to be oriented, Huang et al. [28] propose a second stage, relying on a novel measure of mechanism dependence to individually orient remaining edges.

A consistent yet sparingly explored theme across all of these methods is the *unique* identifiability of the true DAG, rather than an equivalence class, and the impact of the sparsity of changes across environments. Although existing works [29, 57, 62] characterize and identify subsets of the MEC, requirements on the identifiability of the true DAG are unclear. In ICP, it is briefly noted that the method is applicable when the target variable experiences sparse shifts, as long as we assume a maximum degree of sparsity [44, §6.2]. Although not discussed, the performance of methods from Ghassami et al. [13, 14] also depend on sparsity. As mentioned by Huang et al. [28], identifiability requires the invariance of some mechanisms and their pooled PC approach cannot identify edge directions when both adjacent variables change. We will provide clarity on identifiability via sparsity.

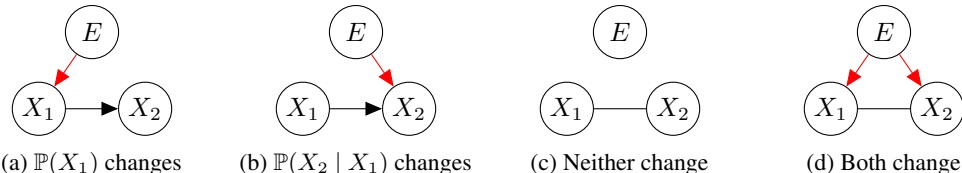

(a) $\mathbb{P}(X_1)$ changes    (b) $\mathbb{P}(X_2 \mid X_1)$ changes    (c) Neither change    (d) Both change

Figure 2: (a, b) Visualizations of the two cases explored in the proof of Prop. 4.1. In both cases, $X_2 \not\perp\!\!\!\perp E$ unconditionally and $X_1 \not\perp\!\!\!\perp E \mid X_2$ directly or by conditioning on the potential collider $X_2$. (c, d) The two other possible situations; neither allow us to orient the edge between $X_1$ and $X_2$.

## 4 Leveraging sparse mechanism changes for causal discovery

The augmented DAG is a powerful tool for understanding the implications of changing causal mechanisms. We build up intuition in the bivariate setting and then consider the general multivariate setting. From now on, we assume causal faithfulness on the augmented CGM.

**Bivariate case.** Given a causal model composed of two associated variables, identifiability of the causal direction requires assumptions such as the functional form or existence of targeted interventions [45, §4.1]. We show how sparsely changing mechanisms also provide identifiability.

**Proposition 4.1** (Both non-causal conditionals change)**.** *Consider the bivariate setting $X_1 \rightarrow X_2$. If either causal mechanism $\mathbb{P}(X_2 \mid X_1)$ or $\mathbb{P}(X_1)$ changes, then both $\mathbb{P}(X_1 \mid X_2)$ and $\mathbb{P}(X_2)$ change.*

**Corollary 4.2** (Bivariate identifiability)**.** *In the setting of Prop. 4.1, if only one causal mechanism changes (sparsity), then the bivariate causal structure is identifiable.*

*Proof.* (Prop. 4.1) We consider each case separately and use a proof by contradiction (of faithfulness).

(i) If $\mathbb{P}(X_1)$ changes (see Fig. 2a), then $G_{\mathbf{X} \cup E}$ contains the edge $E \rightarrow X_1$. If $\mathbb{P}(X_2)$ remained invariant, then $X_2 \perp\!\!\!\perp E$ (unfaithful due to the unblocked path $E \rightarrow X_1 \rightarrow X_2$). If $\mathbb{P}(X_1 \mid X_2)$ remained invariant, then $X_1 \perp\!\!\!\perp E \mid X_2$ (unfaithful due to the direct path $E \rightarrow X_1$).

(ii) If $\mathbb{P}(X_2 \mid X_1)$ changes (see Fig. 2b), then $G_{\mathbf{X} \cup E}$ contains the edge $E \rightarrow X_2$. If $\mathbb{P}(X_2)$ remained invariant, then $X_2 \perp\!\!\!\perp E$ (unfaithful due to the direct path $E \rightarrow X_2$). If $\mathbb{P}(X_1 \mid X_2)$ remained constant, then $X_1 \perp\!\!\!\perp E \mid X_2$ (unfaithful due to the unblocked collider path $E \rightarrow X_2 \leftarrow X_1$). $\quad\square$

*Proof.* (Cor. 4.2) If either causal mechanism changes, by Prop. 4.1 both conditionals in the non-causal factorization change. Hence, the causal structure is the one with only one mechanism change. $\quad\square$

**Multivariate case.** Non-causal conditional distributions of $X_j$ may change across environments even if the causal mechanism $\mathbb{P}(X_j \mid \mathbf{PA}_j)$ does not change. This occurs if the conditioning set leaves open a dependence between $E$ and $X_j$ in $G_{\mathbf{X} \cup E}$.

**Lemma 4.3.** *For any $X_j \in \mathbf{X}$ and set $\mathbf{Z} \subseteq \mathbf{X} \setminus \{X_j\}$, the conditional distribution $\mathbb{P}(X_j \mid \mathbf{Z})$ changes if and only if the following d-connectedness relationship holds:*

$$X_j \not\perp\!\!\!\perp_{G_{\mathbf{X} \cup E}} E \mid \mathbf{Z} \, .$$

The result follows from the Markov property and faithfulness; for all complete proofs, see Appx. B.

Since the Markov equivalence class is relatively easily available and thus often the starting point of open questions in causal discovery, we specify the implications of Lemma 4.3 in this setting. Note that due to the shared skeleton of all DAGs in the equivalence class, the conditioning set for any $X_j$ in any DAG in the MEC is a subset of $X_j$'s true parents and children $\mathbf{PA}_j^G \cup \mathbf{CH}_j^G$ [34, 56].

**Corollary 4.4.** *For any variable $X_j \in \mathbf{X}$ and set $\mathbf{Z} \subseteq (\mathbf{PA}_j^G \cup \mathbf{CH}_j^G)$ in the augmented graph, the conditional distribution $\mathbb{P}(X_j \mid \mathbf{Z})$ changes if and only if at least one of the following holds:*

*(i) $E \rightarrow X_j$ [a direct cause].*

*(ii) $\exists W_{\mathbf{PA}} \in \mathbf{PA}_j^G \setminus \mathbf{Z}$ such that $W_{\mathbf{PA}} \not\perp\!\!\!\perp_{G_{\mathbf{X} \cup E}} E \mid \mathbf{Z}$ [unblocked path to unconditioned parent].*

*(iii) $\exists W_{\mathbf{CH}} \in \mathbf{CH}_j^G \cap \mathbf{Z}$ such that $W_{\mathbf{CH}} \not\perp\!\!\!\perp_{G_{\mathbf{X} \cup E}} E \mid \mathbf{Z} \setminus W_{\mathbf{CH}}$ [unblocked path to conditioned child].*

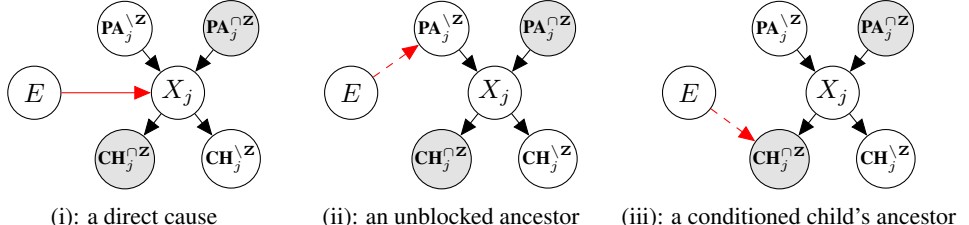

(i): a direct cause     (ii): an unblocked ancestor     (iii): a conditioned child's ancestor

Figure 3: The three possible cases from Cor. 4.4 with a d-connecting path from $E$ to $X_j$, conditioned on a subset of neighbors (colored in grey) in the MEC; these neighbors are a subset of the true parents and children.

*Proof sketch.* These cases are visualized in Fig. 3. The global Markov property and faithfulness assumption allow us to interchange d-connection and a distributional change. In the forward direction, a changing mechanism implies a d-connecting path which is necessarily captured in one of the three cases. In the reverse direction, each case opens a d-connecting path between $X_j$ and $E$. $\qquad\square$

A direct result of changing causal mechanisms and conditional distributions follows.

**Corollary 4.5.** *Let $G^*$ denote the true (unaugmented) DAG and let $G$ be any other DAG over the same variables. For any $X_j \in \mathbf{X}$, a change in $\mathbb{P}(X_j \mid \mathbf{PA}_j^{G^*})$ implies a change in $\mathbb{P}(X_j \mid \mathbf{PA}_j^G)$.*

*Proof.* Under the true causal parents given by $G^*$, a change in mechanism occurs if and only if $E \to X_j$ in $G^*_{\mathbf{X} \cup E}$. So, $X_j$ is d-connected to $E$, no matter the conditioning set $\mathbf{Z} \subset \mathbf{X} \setminus \{X_j\}$. Thus, by Lemma 4.3, $\mathbb{P}(X_j \mid \mathbf{PA}_j^G)$ necessarily changes. $\qquad\square$

# 5   Causal discovery via the Mechanism Shift Score (MSS)

We have established that changing mechanisms provide useful information and by Cor. 4.2 can provide identifiability in the bivariate case under faithfulness and sparse changes. We now study identifiability in more general graphs along with approaches to developing practical estimators. Motivated by the novel discovery of the value of comparing pairwise environments if changes are sparse, we propose the *Mechanism Shift Score (*MSS*)* estimand with useful theoretical guarantees over DAGs in the MEC.

**The MSS estimand.** Given a set $\mathcal{G}$ of candidate DAGs over the same $d$ variables $\mathbf{X} = \{X_1, ..., X_d\}$ with data sampled from $n_\mathcal{E}$ distributions $\mathcal{P} := \{\mathbb{P}_\mathbf{X}^e\}_{e=1}^{n_\mathcal{E}}$, we count the number of changing conditional distributions in a graph $G \in \mathcal{G}$ by defining

$$\text{MSS}_j(G; \mathcal{P}) = \sum_{e' > e}^{n_\mathcal{E}} \mathbb{I}\left[\mathbb{P}^e(X_j | \mathbf{PA}_j^G) \neq \mathbb{P}^{e'}(X_j | \mathbf{PA}_j^G)\right] \quad \text{and} \quad \text{MSS}(G; \mathcal{P}) = \sum_{j=1}^d \text{MSS}_j(G; \mathcal{P}).$$

$\text{MSS}_j(G; \mathcal{P})$ is the number of pairs of environments in which the conditional distribution of $X_j$ in $G$ changes; $\text{MSS}(G; \mathcal{P})$ is the total number of changes across all variables and pairs of environments according to the Markov factorization implied by $G$. It follows from Cor. 4.5 that the true DAG $G^*$ minimizes (not necessarily uniquely) the number of changing conditionals among all DAGs.

**Proposition 5.1.** *Let $G^*$ be the true DAG in the set $\mathcal{G}$ of DAGs. Then for all $G \in \mathcal{G}$ and $j \in \{1, \dots, d\}$*

$$\text{MSS}_j(G^*; \mathcal{P}) \leq \text{MSS}_j(G; \mathcal{P}) \qquad \text{and thus} \qquad \text{MSS}(G^*; \mathcal{P}) \leq \text{MSS}(G; \mathcal{P}).$$

*Proof.* By Cor. 4.5, any change in $\mathbb{P}(X_j \mid \mathbf{PA}_j^{G^*})$ implies a change in $\mathbb{P}(X_j \mid \mathbf{PA}_j^G)$ for any other DAG $G$. Thus, any change counted by $\text{MSS}_j$ on the true DAG will also be detected in every other DAG and so both lower bounds hold. $\qquad\square$

Prop. 5.1 can be viewed as the generalization of the Principle of Minimal Changes [14], allowing for mechanism changes beyond the parametric restrictions of a linear model. Identifiability, however, requires us to establish a discerning aspect of the true structure. Recall that the Markov equivalence class $\mathcal{G}_{\text{MEC}}$ is identifiable. We define the subset

$$\mathcal{G}_{\text{MEC}}^{\min} := \underset{G \in \mathcal{G}_{\text{MEC}}}{\arg\min} \text{MSS}(G; \mathcal{P})$$

of DAGs with minimum MSS, here defined as the number of mechanism shifts across environments.

It turns out that $\mathcal{G}_{\text{MEC}}^{\min}$ is a $\Psi$-MEC with respect to the true but unknown intervention targets $\{\mathcal{I}^e\}_e$ as characterized by Jaber et al. [29]. We defer further details and the proof of this correspondence to Appx. B and focus our results on unique identifiability of the true DAG here.

In practice, we may employ any generic conditional test for change in mechanism or choose a "softer" score (e.g., based on $p$-values) to quantify changes along a continuous spectrum. This is a similar idea to that of Brouillard et al. [4] who propose a multi-environment likelihood-based approach.

Prop. 5.1 implies that $G^* \in \mathcal{G}_{\text{MEC}}^{\min}$. Using probabilistic assumptions based on the idea of sparse changes, we show that, given enough environments, the causal parents and full DAG are identifiable.

**Lemma 5.2** (Identifiability of causal parents). *Let $G^*$ be the true DAG in the MEC $\mathcal{G}_{\text{MEC}}$ and $\rho_i$ the probability that the causal mechanism of $X_i$ is different across any two environments. Under Asms. 2.2 to 2.4 and 2.6, for any $j \in \{1, \ldots, d\}$, graph $G \in \mathcal{G}_{\text{MEC}}$ such that $\boldsymbol{PA}_j^{G^*} \neq \boldsymbol{PA}_j^G$, and lower and upper bounds on the shift probabilities $\rho_i^{\text{LB}} \leq \rho_i \leq \rho_i^{\text{UB}}$ for all $i$, we have that*

$$\Pr[\text{MSS}_j(G^*; \mathcal{P}) < \text{MSS}_j(G; \mathcal{P})] \geq 1 - \left(1 - (1 - \rho_j^{\text{UB}}) \min_i \rho_i^{\text{LB}}\right)^{\lfloor n_{\mathcal{E}}/2 \rfloor}.$$

*Proof sketch.* From Cor. 4.5, we know: $\text{MSS}_j(G^*; \mathcal{P}) \leq \text{MSS}_j(G; \mathcal{P})$. We use the shared skeleton property of all DAGs in a MEC and Cor. 4.4, for which only case (i) admits a changing mechanism in the true DAG. Based on the ICM principle, we examine sufficient conditions in a pair of environments and establish a probabilistic upper bound on all pairs. □

Note that this bound is independent of the other DAG $G$, so long as $G$ is in the MEC and has a different set of causal parents. As a special case of Lemma 5.2, invariance of the mechanism of $X_j$ implies $\rho_j^{\text{UB}} = 0$ and hence provides a bound relevant to invariant causal prediction [23, 44]. Building off of Lemma 5.2, we can provide a probabilistic bound on identifiability of the whole graph.

**Theorem 5.3** (Identifiability of the graph). *Let $G^*$ be the true DAG in the MEC $\mathcal{G}_{\text{MEC}}$ and $\rho_j$ the probability that the causal mechanism of $X_j$ is different across any two environments. Under assumptions 2.2, 2.3, 2.4, and 2.6, and bounds $\rho_i^{\text{LB}} \leq \rho_i \leq \rho_i^{\text{UB}}$ for all $i$, we have that*

$$\Pr[\mathcal{G}_{\text{MEC}}^{\min} = \{G^*\}] \geq 1 - |\mathcal{G}_{\text{MEC}}| \left(1 - (1 - \min_i \rho_i^{\text{UB}}) \min_i \rho_i^{\text{LB}}\right)^{\lfloor n_{\mathcal{E}}/2 \rfloor}.$$

*Proof sketch.* From Prop. 5.1, $G^*$ is always in $G_{\text{MEC}}^{\min}$. For each DAG, we use Lemma 5.2 to bound the probability that all mechanisms exhibit the same number of changes. Then we apply the union bound to establish an upper bound across all DAGs. □

**Corollary 5.4.** *If $\rho_i$ is bounded away from 0 and 1 for all $i$, (a probabilistic form of Asm. 2.7),*

$$\Pr[\mathcal{G}_{\text{MEC}}^{\min} = \{G^*\}] \to 1 \quad as \quad n_{\mathcal{E}} \to \infty$$

*That is, with enough environments we can recover the true DAG from the Markov equivalence class.*

*Proof.* The assumption of bounded probability implies that $\rho_i^{\text{UB}} < 1$ and $\rho_i^{\text{LB}} > 0$ for all $i$. Hence, by the rate established in Thm. 5.3, identifiability is achieved in the limit. □

**The MSS estimator.** An empirical MSS estimator tests if two conditional distributions change across two environments. This can be done using conditional independence tests or equality of distribution tests [39]. Under parametric assumptions, models may be fit for each mechanism, and these parameters can then be tested across environments [13, 14, 44]. Heinze-Deml et al. [23] provided a comprehensive study of such tests and their power for ICP. More recent but less studied work by Park et al. [40] has provided a kernel-based approach with strong guarantees. In practice care must be taken when using equality of conditional distribution tests, especially if any of their assumptions are violated [53].

**Computational complexity.** The score function $\text{MSS}(G; \mathcal{P})$ is *decomposable* [19] in that it is the sum of local scores $\text{MSS}_j(G; \mathcal{P})$. In the most naïve approach, each mechanism in each of $|\mathcal{G}_{\text{MEC}}|$ DAGs must be tested across all pairwise environments, on the overall order of $O(|\mathcal{G}_{\text{MEC}}|dn_{\mathcal{E}}^2)$, without accounting for the complexity of the statistical test. This can be slow, but if the test runtime scales with sample size $n$ faster than $O(n^2)$, then pairwise tests may actually be faster than pooling the data. In practice, the decomposable property permits a speedup: since many mechanisms will be shared across DAGs, we can test each unique mechanism and then select the results relevant for each DAG. Experimentally, we find the main bottlenecks to be large sample sizes and numbers of environments.

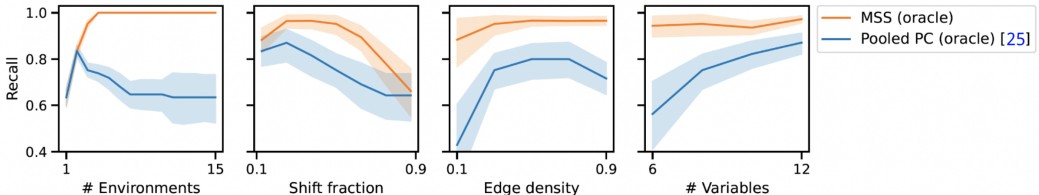

Figure 4: **Oracle rates match the theory.** From left to right (a-d): (a) With sufficiently many environments, our MSS approach learns the true DAG while pooled PC recovers only the original MEC. (b) Pairwise comparisons are less beneficial when shifts are extremely sparse or dense. Differences across the two methods are particularly pronounced both in (c) sparse and dense DAGs, as well as in (d) smaller DAGs. Shaded regions denote 95% confidence intervals calculated from bootstrapped data.

## 6 Structure learning experiments

Having established the theoretic value of the MSS estimand, we seek to understand *empirically*: (i) how the oracle MSS and pooled PC approaches compare across experimental settings, (ii) which MSS estimators perform best, and (iii) how the MSS compares to related approaches.[4] For ease of comparison, we adapt the simulation setup of Huang et al. [28]. Random DAGs are sampled using an Erdős-Rènyi model [12] in which each edge has some fixed probability of existing (the *edge density*). In each environment, a random set of variables experience a mechanism change according to a fixed number or fraction of shifts. Each variable $j$ in environment $e$ has a randomly sampled mechanism

$$X_j^e := \sum_{i \in \mathbf{PA}_j} b_{ji}^e f_{ji}(X_i^e) + \sigma_j^e \epsilon_j^e \qquad (6.1)$$

where $b_{ji} \sim \mathcal{U}(0.5, 2.5)$, $\sigma_j^e \sim \mathcal{U}(1, 3)$, and $\epsilon_j^e \sim \mathcal{N}(0, 1)$ or $\mathcal{U}(1, 3)$ with equal probability. The functions $f_{ji}$ are selected uniformly at random from $\{x^2, x^3, \tanh, \text{sinc}\}$. Mechanisms in an unobserved baseline environment are sampled and $\sigma_{ji} = 1$ is fixed. Each observed environment inherits the baseline distributions and mechanisms shifts are resampled per (6.1).

We evaluate the quality of an estimated CPDAG against the true DAG via *precision* and *recall* [13, 14, 28]. Precision is the fraction of *directed* edges in the CPDAG which are correctly oriented. Recall is the fraction of *all* edges in the CPDAG which are oriented. Thus, the true DAG has perfect precision and recall. Since all methods start from the MEC, there are no incorrect edges, only incorrect orientations. For oracle methods, the precision is perfect and so we only report the recall.

**Oracle MSS rates match the theory.** Having theory on learning rate bounds under sparsity, we now seek to understand how the empirical performance of MSS and pooled PC depends on a variety of graph and sparsity settings, both which our theory does and does not address. We consider random DAGs over 6 variables with edge density 0.3. Five environments are sampled, in each of which half of the mechanisms shift. In Fig. 4, we hold all of these settings fixed and vary one at a time across 50 repetitions, comparing the recall of the two methods.[5] Precision is always perfect under the oracle test.

In the first two plots, the empirical results match what the theory predicts. First, per Cor. 5.4, with more environments MSS learns the entire graph while pooled PC learns nothing but the original MEC. Second, per Thm. 5.3, the learning rate decreases when shifts are either uncommon or frequent. Furthermore, we see that differences between the two approaches are accentuated in sparse and dense DAGs, as well as in smaller DAGs. Note that sparsity is a fixed fraction of the variables, and hence larger DAGs experience more shifts in absolute terms. See Appx. D for additional oracle experiments.

**MSS performance depends on the chosen estimator.** Next, we study how the choice of estimator and type of score affects performance. We use two popular conditional independence tests, the Fisher-Z partial correlation test [31] and the Kernel Conditional Independence (KCI) test [65], as well as the invariant residual test using a generalized additive model (GAM), a top-performing ICP method [23].[6] Each detects a change if the test $p$-value is less than $\alpha := 0.05/d$, a Bonferroni correction to bound the false positive rate for each scored DAG. Although the theory pertains to counting shifts, in settings where a "hard" hypothesis test has low power it may be of more value

---

[4]All code and experiments are available at https://github.com/rflperry/sparse_shift

[5]Oracle methods used code from the *causaldag* package [3-Clause BSD license].

[6]KCI and Fisher-Z are implemented by the *causal-learn* package [GNU General Public License].

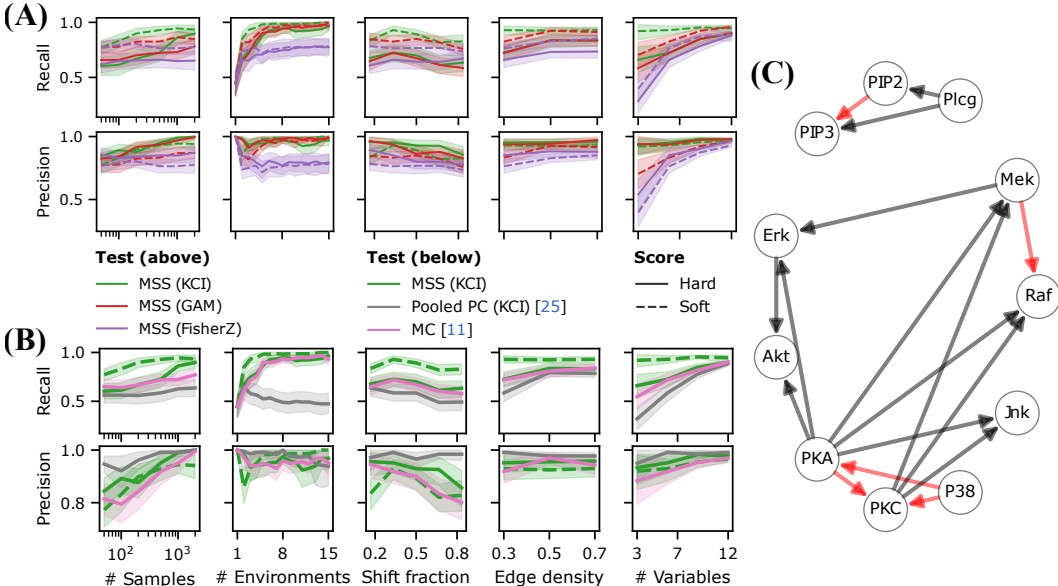

Figure 5: **(A)** Nonparametric hypothesis tests perform well in nonlinear simulations, and soft scores succeed. Notably, recall converges with increasing environments. KCI appears to best balance high recall and precision. **(B)** In simulation, pairwise approaches improve with more environments, unlike Pooled PC. Although the parametric MC works surprisingly well across settings, the nonparametric MSS with KCI has superior precision and recall. **(C)** MSS (KCI) edge orientations mostly match the real data Sachs network [49]. Non-matching edges (red) are posited to be involved in cycles and of ambiguous orientation in the literature.

to use a "softer" score (e.g., based on $p$-values) to quantify changes along a continuous spectrum. We propose the following "soft" score (see Appx. C for further details):

$$\widehat{\mathrm{MSS}}_j(G;\mathcal{D}) = \sum_{e=1,e'>e}^{n_\mathcal{E}} \left[ 1 - p\text{-value}\left( \mathbb{P}^e(X_j \mid \mathbf{PA}_j^G) \neq \mathbb{P}^{e'}(X_j \mid \mathbf{PA}_j^G) \right) \right].$$

DAGs are generated on six variables with edge density 0.3. Three environments are sampled, with 500 samples and two mechanism shifts per environment. We vary each variable while holding the others fixed and compare results across 50 repetitions.

As seen in Fig. 5(A), the Fisher-Z test performs noticeably worse, presumably due to unmet parametric assumptions, while the nonparametric approaches do well, particularly at higher sample sizes. As with the oracle, the true DAG is recovered with enough environments. The "soft" versions achieve higher recall, since there is a unique minima, at the cost of worse precision (see Appx. C.2 for greater discussion). The "soft" KCI seems to be best, suggesting that KCI models the data the best. In practice, it is crucial for a method to model the data well for the $p$-value to be valid [53].

**MSS compares favorably against other methods.** Next, we compare MSS to relevant existing methods: the *Minimal changes (MC)* approach [14] tests pairs of environments for changes in the parameters of a linear model, and Huang et al. [28] provide a nonparametric version—a two-stage approach which first uses PC with the KCI test on the pooled data. We investigate if pooling data loses information under empirical tests, and how the nonparametric pairwise MSS test compares.

In Fig. 5(B), we compare these approaches in the previously studied experimental setting. Pooled PC has quite high precision (as it pools all of the data), but suffers from lower recall, especially with more environments when recall is no better than the base MEC. In contrast, "hard" KCI has higher recall at the cost of some precision when individual environments have few samples. The parametric MC works surprisingly well, yet slightly worse than KCI. Overall, MSS seems to combine the best of both approaches: the value of pairwise comparisons from MC with the flexibility of incorporating various nonparametric estimators. Additional experiments in Appx. D confirm this in the bivariate case.

**Protein network discovery.** As an illustration of our method in practice, we conduct a case-study application of MSS for causal discovery on a well-studied cytometry dataset [49] consisting of 9 experimental environments of 11 cellular proteins. Starting from the Sachs MEC, we apply the MSS using the KCI test, which appears to perform the best among plug-in estimators for MSS in our simulations.

The DAG which minimizes the MSS is the unique minimizer and is visualized in Fig. 5(C). Learned edge orientations mostly match the Sachs network [49]. Non-matching edges, shown in red, are posited to be involved in cycles and of ambiguous orientation in the literature: PIP2 → PIP3 is known to be a cyclical relation [47, 49], Mek → Raf is indeed found by many other methods [8, 38, 47], and although there is not a detailed discussion of the PKA, PKC, P38 triangle, there is ambiguity in the edge directions among approaches [8, 38, 47], as we discuss in more detail in Appx. D.3.

# 7 Discussion

**Sparse shifts as a relaxation of the i.i.d. assumption.** Distribution shifts are a common violation of the i.i.d. assumption and a problematic source of error in practice. It has been argued that the issue of robustness to natural shifts is connected to causality [1, 32, 44, 50, 60, 61, 64]. We have shown that viewing a shift in distribution through the causal factorization permits a useful relaxation of the i.i.d. assumption, facilitating causal discovery: if there are no shifts (i.i.d.), we can only identify equivalence classes. If shifts are unrestricted, we cannot meaningfully transfer across distributions [50], and ideas such as Simon's invariance criterion [25] and ICP [44] do not help. However, if shifts occur *sparsely*—as we formalize—we can provably use this as a learning signal to infer the causal structure.

**The Mechanism Shift Score (MSS) and prior methods.** The MSS framework extends previous causal discovery work limited to linear mechanisms [14], just as nonlinear ICP [23] extended the initially limited ICP approach [44] beyond linear models. Huang et al. [28] do provide a useful nonparametric approach, but as we have demonstrated, by pooling all of the data in their first stage they lose out on the value of sparse shifts in identifying the true graph. The MSS is both nonparametric and explicitly leverages sparsity through pairwise comparisons. Our accompanying graph-theoretic analysis proves why pairwise comparisons are actually useful, and our learning rates apply to these previous works to provide insight on the role of sparsity.

**Beyond causal discovery.** Once the causal graph is known, conditional distributions and hence an entire causal graphical model can be learned. This is harder than learning a statistical model, but has various advantages, especially when distributions shift, as they do in reality. E.g., we may be able to use such a model for causal reasoning, i.e., estimating a certain causal effect. We also expect that MSS can serve as a useful inductive bias for causal representation learning, similar to how invariant prediction [44, 50] inspired invariant risk minimization [32]; recent work has started to explore this [34, 36].

**Empirical performance and limitations.** We infer causal structure through a flexible score-based method; as empirically demonstrated, strong results can be obtained by multiple estimators when the assumption of sparsity is met. This requires hypothesis tests which can accurately obtain the full MEC as a starting point and subsequently test for distribution changes. Among additional assumptions including the stringent pseudo-causal sufficiency, sparsity is necessary and yet not easily verifiable; SMS is indeed a *hypothesis* regarding causal systems, not a fact. We conjecture that under an oracle, the "hard" MSS is equivalent to the PC algorithm pooled pairwise across environments. It is worth noting that the "hard" approach may still be useful under dense changes if only a sparse number of them are large enough to be *empirically discernible*. Thus the empirical method can actually outperform the oracle baseline and be useful even if the assumption of sparsity is unmet. We also assume that a partition into environments is known, but environment inference techniques may help relax this [6].

**Outlook and conclusion.** Imagining causal models on an axis of complexity, from the microscopic physical laws of nature to a simplified set of variables and relationships, we transition from a system with no mechanism shifts (effectively a dynamical system) to a system in which all mechanisms shift (due to many unmeasured causes). In the middle, we posit only a sparse number of shifts to be empirically discernible. While *all (causal) models are wrong*, the one which is most invariant to shifts may be the best candidate for supporting robust and transferable inference.

## Acknowledgments and Disclosure of Funding

We thank Jonas Kübler, Junhyung Park, Krikamol Muandet, Luigi Gresele, the Tübingen causality team, and the anonymous reviewers for helpful comments. This work was supported by the German Federal Ministry of Education and Research (BMBF): Tübingen AI Center, FKZ: 01IS18039A, 01IS18039B; and by the Machine Learning Cluster of Excellence, EXC number 2064/1 – Project number 390727645. Ronan Perry was supported by a Fulbright Germany research fellowship.

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
