# OpenReview forum: "Causal Discovery in Heterogeneous Environments Under the Sparse Mechanism Shift Hypothesis"
_NeurIPS.cc/2022/Conference — NeurIPS 2022 Accept_

### Official Review · Reviewer_qyjw · 2022-07-02

**Rating:** 6
**Confidence:** 3
**Soundness:** 3 good
**Presentation:** 3 good
**Contribution:** 3 good

**Summary:**

This paper proposes a score-based approach for causal discovery (Mechanism Shift Score) using non-iid data from multiple environments under spare mechanism shifts. It provides theoretical analysis for the proposed approach, as well as empirical experiments.

**Questions:**

-In evaluation, why do you use a data generation mechanism where the effect of parents and noise is additive? Does this method not work for a more general data generation mechanism? Why?

-Comparison with "https://arxiv.org/abs/1910.01075" would be interesting.

-Why don't you show F-1 scores for evaluation?

Line 285: rates' instead of rates
Line 234: Corollary 4.4 instead of Lemma 4.3

**Limitations:**

Yes

**Strengths And Weaknesses:**

Although the Sparse Mechanism Shift hypothesis and its application for causal discovery is not new, this paper provides a sound method for using it by designing a score and analyzing it theoretically.

Pros:
-The presentation is clear and easy to follow.
-The problem is significant in many areas of science.

Cons:
-Evaluation in the main body is limited to only a synthetic data generation mechanism.

---

> ### Author Response · Authors · 2022-08-02
> **Response to Reviewer `qyjw`**
>
> Thank you for your valuable feedback. We have attempted to quote each of your raised points and provide a satisfactory answer and/or revision.
>
> > "Although the Sparse Mechanism Shift hypothesis and its application for causal discovery is not new, this paper provides a sound method for using it."
>
> We appreciate the positive feedback, but would like to push back slightly on the statement that the application of the SMS hypothesis for causal discovery is not new. Although certainly being *implicitly* related to numerous works, we are unaware of causal discovery work which *explicitly* assumes it in theoretical work or studies its impact on learning.  We would be happy to see concrete references in which the SMS is explicitly leveraged for structure identifiability, and to include them in our related works section.
>
> > "Evaluation in the main body is limited to only a synthetic data generation mechanism."
>
> Indeed, there is a lack of standardized and trustworthy real world benchmarks for causal discovery, especially in the multi-environment setting; this is even more so the case when we wish to study the impact of the sparse mechanism shift hypothesis in particular. Following your comment, we have shifted portions of our study on the Sachs cytometry dataset from the appendix to the main body in order to better highlight this case study.
>
> > "In evaluation, why do you use a data generation mechanism where the effect of parents and noise is additive? Does this method not work for a more general data generation mechanism? Why?"
>
> For ease of comparison, we adopted a previously published experimental setting, which happened to only use additive noise. As with many other causal discovery methods including Invariant Causal Prediction [S4] and Huang et al. 2020 [S5] who we compare to, our MSS method relies on testing for the equality of conditional distributions for which a conditional independence test is a common approach; certainly some tests are less suited for certain distributions. However, our MS approach permits any test, including nonparametric ones, unlike an approach such as the MC [R6] which relies on purely linear models. Overall, the specifics of the distribution are separate from our focus on the SMS hypothesis. However, for added completeness we have complemented the power study in Appendix Figure 12 with Appendix Figure 13 which presents a multiplicative noise model. The results and discussion there reflects our points above.
>
> > "Comparison with Ke et al. 2019 would be interesting."
>
> We have included a citation of Ke et al. (2019) [S2] as it is indeed related in that it performs causal discovery in a multi-environment setting with unknown intervention targets. Yet, in its present form, it is only applicable to discrete random variables. As suggested by `Rbq8`, the method of Brouillard et al. 2020 [S1] would be a more useful comparison, but unfortunately they do not provide experiments or code in our setting of soft interventions with unknown targets.
>
> > "Why don't you show F-1 scores for evaluation?"
>
> Thank you for the suggestion. In the revised manuscript, we have added F1 scores to Figs. 5 and 6. Although the F1 score unifies precision and recall and is thus a summary measure, we believe that it is also important to view both separately in order to not obfuscate relevant details. For instance, as discussed with `m9g9` with respect to Figure 6, Pooled PC is quite precise, since it has access to all of the data, at the cost of significantly lower recall.
>
> [S1] Brouillard, P., Lachapelle, S., Lacoste, A., Lacoste-Julien, S., \& Drouin, A. (2020). Differentiable causal discovery from interventional data. Advances in Neural Information Processing Systems, 33, 21865-21877.
>
> [S2] N. R. Ke, O. Bilaniuk, A. Goyal, S. Bauer, H. Larochelle, C. Pal, and Y. Bengio. Learning neural causal models from unknown interventions. arXiv preprint arXiv:1910.01075, 2019.

---

> > ### Comment · Reviewer_qyjw · 2022-08-08
> > **After rebuttal**
> >
> > For the application of SMS in causal discovery, I recall https://arxiv.org/abs/1901.10912, which also has a discussion and investigation of the bivariate case.
> >
> > For evaluation in the main body, I think it makes more sense to use a setting where observational data by itself is insufficient for causal discovery rather than an additive noise setting.
> >
> > I increased my score.

---

> > > ### Author Response · Authors · 2022-08-09
> > > **Discussion follow-up**
> > >
> > > Thank you for the reference; we are familiar with the work, and while it focuses on a somewhat different setting (fast adaptation to covariate shift in a continual/meta learning setting as a causal learning signal), it indeed highlights the broader value of the SMS hypothesis, which we similarly exploit, and we will cite it in the next revision.
> > >
> > > Although our identifiability results are agnostic to the underlying data generating process, we acknowledge your point on the experimental setup (re: additive noise). We adopted a previously published setup to provide a comprehensive comparison with existing methods, but the more general case may also be informative. We will include a note on this and a pointer to the multiplicative noise experiments in the Appendix in the next revision. To add one more point on this matter: while ANMs are indeed observationally identifiable with infinite data, it is not clear that meaningful and reliable results are obtainable in a multi-variate setting with only hundred(s) of samples; investigating alternative routes to identifiability in a different setup with several environments but few samples per environment may thus still be valuable even for the ANM class.
> > >
> > > Thanks again for engaging in the discussion and increasing your score.

---

### Official Review · Reviewer_m9g9 · 2022-07-10

**Rating:** 6
**Confidence:** 4
**Soundness:** 3 good
**Presentation:** 4 excellent
**Contribution:** 3 good

**Summary:**

This paper focuses on the identifiability of causal graphs up to CPDAG in heterogeneous environments. Specifically, they explore the assumption of minimal changes in causal conditionals across different environments. This assumption of sparse mechanism shift is a non-parametric extension of the Principle of Minimal Changes [1]. With this as the hypothesis, a score-based method that identifies the entire causal structures with high probability has been proposed.

-------

[1] Amir Emad Ghassami, Negar Kiyavash, Biwei Huang, and Kun Zhang. Multi-domain Causal
Structure Learning in Linear Systems. In Advances in Neural Information Processing Systems,
volume 31. Curran Associates, Inc., 2018.

**Questions:**

Please let me know if the points/questions listed as the cons are due to my misunderstanding of the manuscript.

**Limitations:**

It seems that there is no obvious discussion of limitations. Please let me know if I missed anything.

**Strengths And Weaknesses:**

Pros:

1. The idea of exploring sparse mechanism shifts in a non-parametric setting is original and of interest to the community.
2. The theoretical results and assumptions have been rigorously stated.
3. The background and related works have been well introduced, which make the general picture clearer.
4. The writing is very clear.

Cons:

1. From Fig. 6, one could observe that MSS (KCI), which is based on the proposed score, doesn't outperform other baselines w.r.t. precision. More discussion about the reason would be appreciated.

2. I didn't find a clear statement/discussion of the limitation of the model. In the checklist, the limitation part has been pointed to the third paragraph of Sec. 7. However, it seems that no limitation has been clearly discussed in this Section. The most related part is the paragraph of 'Empirical performance and the validity of the sparse mechanism shift hypothesis.', but I still fail to see the discussion of limitations. Perhaps a direct discussion of them could be helpful?

3. Regarding [2], it has been mentioned in the manuscript that, compared to the proposed method, the disadvantage of [2] is that it is not suited for sparsely-detectable changes. Since the sparse change is basically the assumption made in this manuscript but not in [2], I'm wondering whether it is a valid/fair point in general cases.



-----------------

[2] Biwei Huang, Kun Zhang, Jiji Zhang, Joseph Ramsey, Ruben Sanchez-Romero, Clark Glymour, and Bernhard Schölkopf. Causal discovery from heterogeneous/nonstationary data. Journal of Machine Learning Research, 21(89):1–53, 2020.

---

> ### Author Response · Authors · 2022-08-02
> **Response to Reviewer `m9g9`**
>
> Thank you for your valuable feedback. We have attempted to quote each of your raised points and provide a satisfactory answer and/or revision.
>
> > "From Fig. 6, one could observe that MSS (KCI) ... doesn't outperform other baselines w.r.t. precision. More discussion about the reason would be appreciated."
>
> Indeed, this is a useful point of clarity, and we have elaborated our discussion in line 335. Note that recall counts the fraction of undirected edges in the MEC which we orient, while precision counts how many of those orientations are correct. Thus, predicting the MEC has perfect precision but the worst recall. In Fig. 6 we see that Pooled PC [S2] has high precision, by virtue of *pooling all* of the data, but has recall shrinking with additional environments. In contrast, MSS has slightly lower precision but much higher recall. Although MSS and MC have similar precision, MSS dominates in recall.
>
> > "I didn't find a clear statement/discussion of the limitation of the model."
>
> We apologize that this was not clear. Our various assumptions are listed throughout, and experiments include regimes in which the MSS faces limitations. As a summary, the following statement has been added to the discussion (line 381):
>
> *"We infer causal structure through a flexible score-based method and, as empirically demonstrated, strong results can be obtained by multiple estimators when the assumption of sparsity is met. This naturally requires empirical hypothesis tests which can accurately obtain the full MEC as a starting point and subseqeuntly test for distribution changes. Among various additional assumptions including the stringent pseudo-causal sufficiency, the assumption of sparsity is necessary and yet not easily verifiable; the SMS is indeed a hypothesis regarding causal systems, not a fact."*
>
> > "Regarding Huang et al. 2020 [S2], it has been mentioned in the manuscript that, compared to the proposed method, the disadvantage of [S2] is that it is not suited for sparsely-detectable changes. Since the sparse change is basically the assumption made in this manuscript but not in [S2], I'm wondering whether it is a valid/fair point in general cases."
>
> Actually, Huang et al. 2020 [S2] is not suited for dense changes and already *relies on sparse changes* in order to learn more than the MEC. Broadly, as we discuss on line 157, [S2] belongs to a family of works implicitly relying on SMS. The issue we highlight in Figs. 1, 4, and 6 is that even under sparse changes within individual environments, their method pools all environments and hence tends towards dense changes overall. In contrast, our pairwise approach preserves sparsity and our theory explicitly shows how the SMS enables identifiability of the true DAG.
>
> [S1] Amir Emad Ghassami, Negar Kiyavash, Biwei Huang, and Kun Zhang. Multi-domain Causal Structure Learning in Linear Systems. In Advances in Neural Information Processing Systems, volume 31. Curran Associates, Inc., 2018.
>
> [S2] Biwei Huang, Kun Zhang, Jiji Zhang, Joseph Ramsey, Ruben Sanchez-Romero, Clark Glymour, and Bernhard Schölkopf. Causal discovery from heterogeneous/nonstationary data. Journal of Machine Learning Research, 21(89):1–53, 2020.

---

> > ### Comment · Reviewer_m9g9 · 2022-08-07
> > **After rebuttal**
> >
> > Thanks for the clarification. I tend to keep my score and suggest an acceptance.

---

### Official Review · Reviewer_Rbq8 · 2022-07-10

**Rating:** 5
**Confidence:** 5
**Soundness:** 3 good
**Presentation:** 4 excellent
**Contribution:** 2 fair

**Summary:**

In this paper, the authors proposed a causal structure learning algorithm based on the "sparse mechanism shift hypothesis" which says that distribution shifts across environments occur due to a small number of changes in the causal conditions. They introduced the "Mechanism Shift Score" which is the number of causal conditions changing across all pairs of environments, showing that the true causal graph minimizes this score. Furthermore, they showed that with enough changes across environments, the true causal graph can be identified.

**Questions:**

1- The class of graphs that are identified by the proposed method should be characterized. Is it equivalent to I-Markov or $\Psi$-Markov class? I think the results in Theorem 5.3 is fairly straightforward.

2- I think one of the main issues of the proposed approach (and also the one in [25]) is that we need many environments to perform accurate conditional independence tests.

3- Can this work be extended to the case that we have latent confounders? For instance, the result in [Jaber et al. 2020] allows the latent confounders.

4- In the experiments, it is better to compare different methods (including the missing references) in terms of F1 score and SHD.

**Limitations:**

I think it is a theoretical work and the authors clearly mentioned the assumptions in their work.

**Strengths And Weaknesses:**

In this paper, the same augmented graph in [25] is constructed. However, instead of pooling all data, pairwise comparison among environments is considered which might give more information as shown in Figure 1. The paper is generally well written and the assumptions are clearly mentioned. However, it seems that some of the related works are missed. For instance, in [Brouillard et al. 2020], a score function for the setting of multi-environment with unknown target interventions has been proposed where the true causal graph can be identified up to an I-Markov equivalence class. There, it is assumed that the distribution of one of the environments is observational and there is no intervention on any variable, and we are aware of which environment has this property. Later, [Jaber et al. 2020] relaxed this assumption and defined an equivalence class for the setting of multi-environment with unknown target interventions (called $\Psi$-Markov equivalence) and characterized this class. Furthermore, they proposed an algorithm that is sound and complete. I believe the submitted paper should compare the results here both theoretically (whether it can recover causal structures that cannot be identified with previous work) and also experimentally.


[Brouillard et al. 2020] Brouillard, P., Lachapelle, S., Lacoste, A., Lacoste-Julien, S., & Drouin, A. (2020). Differentiable causal discovery from interventional data. Advances in Neural Information Processing Systems, 33, 21865-21877.

[Jaber et al. 2020] Jaber, A., Kocaoglu, M., Shanmugam, K., & Bareinboim, E. (2020). Causal discovery from soft interventions with unknown targets: Characterization and learning. Advances in neural information processing systems, 33, 9551-9561.

---

> ### Author Response · Authors · 2022-08-02
> **Response to Reviewer `Rbq8` (part 1/2)**
>
> Thank you for your valuable feedback and knowledge of related literature. We have attempted to quote each of your raised points and provide a satisfactory answer and/or revision.
>
> > "The class of graphs that are identified by the proposed method should be characterized. Is it equivalent to $\mathcal{I}$-Markov or $\Psi$-Markov class?"
>
> Thank you for providing these equivalence class ideas and related references. Indeed, under an oracle test and without latent variables, the MSS solution set is a $\Psi$-Markov equivalence class (MEC) [S2] of DAGs with respect to the true but unknown intervention targets. We refer to line 233 and appendix Proposition B.1 in the revised manuscript. Although our setting does not concern latent variables, as pointed out by Jaber et al. (2020) [S2] the $\Psi$-MEC is still a generalization of the $\mathcal{I}$-MEC [S1, S3] since no observational environment needs to be defined, and so can yield a smaller equivalence class.
>
> > "some of the related works are missed. For instance... [Brouillard et al. 2020]... [Jaber et al. 2020]"
>
> Thank you for pointing us to these useful references. We have updated the manuscript to include discussions and comparisons at lines 141, 158, and 233. In summary:
>
> - **Brouillard et al. (2020)**: The authors provide a score-based approach to learning a DAG in the $\mathcal{I}$-MEC from hard/soft and known/unknown interventions. It is similar in spirit to our soft score MSS version, but  they model the data distribution and use a log-likelihood score. Theoretically, they show that their method recovers the $\mathcal{I}$-MEC, but since this restrictively requires defining a known observational environment, we study the generalized $\Psi$-MEC [S2] as suggested. Experimentally, while the DCDI method of Brouillard et al. (2020) appears applicable to our setting (soft and unknown interventions), their experimental work does not address this setting and their code does not support it.
>
> - **Jaber et al. (2020)**: The authors provide a sound and complete algorithm for learning the $\Psi$-MEC from soft and unknown interventions, even in the presence of unmeasured latent variables. Theoretically, they define the $\Psi$-MEC, and we prove that our MSS estimator similarly recovers this equivalence class. Under this framing, our main theoretical results show when the $\Psi$-MEC shrinks to just the true DAG. Experimentally, there is unfortunately no implemented algorithm to compare to as the work was purely theoretical and only considered the population (infinite data) setting.
>
> With respect to both works, we would like to note that while such methods provably recover certain equivalence classes (see line 157), there is a lack of study on when the equivalence class is solely the true DAG (unique structure identifiability). Our theory fits into the broader literature by clearly demonstrating that the SMS hypothesis relaxes the i.i.d. assumption (under which only the MEC can be learned) and provides identifiability given sufficiently many environments.
>
> > "I think one of the main issues of the proposed approach (and also the one in [25]) is that we need many environments to perform accurate conditional independence tests."
>
> Although more environments *can* indeed be useful for [25], high precision is generally dependent on large sample sizes, which are attainable even with just two environments. Specifically, we are interested in testing $P^1(X\vert Y) = P^2(X\vert Y)$, i.e., whether two conditional distributions are equal. This can be achieved through both conditional two-sample tests *and* conditional independence tests. With such tests, sample size is typically the dominant factor in ensuring high power. As we discuss in our paper (line 262), Heinze-Deml et al. (2018) provide an extensive analysis on the power of various tests; see Appendix Figure 12 as well for comparisons of various tests' power and precision in a bivariate system. In our simulations, the Kernel Conditional Independence (KCI) test was consistently the best performing even in just two environments.
>
> > "Can this work be extended to the case that we have latent confounders? For instance, the result in [Jaber et al. 2020] allows the latent confounders."
>
> Without latent confounders, we have shown that our approach recovers the $\Psi$-MEC [S2]. Thus, it seems reasonable that the MSS estimator could be similarly  extended to deal with generic latent confounders in future work by using ideas from Jaber et al. (2020). However, such an extension is beyond the scope of our current work, whose focus lies on the identifiability provided by the Sparse Mechansism Shift hypothesis and experiments in unconfounded settings.

---

> > ### Author Response · Authors · 2022-08-02
> > **Response to Reviewer `Rbq8` (part 2/2)**
> >
> > > "In the experiments, it is better to compare different methods (including the missing references) in terms of F1 score and SHD."
> >
> > Thank you for the suggestion. In the revised manuscript, we have added F1 scores to Figs. 5 and 6. Although the F1 score unifies precision and recall and is thus a summary measure, we believe that it is also important to view both separately in order to not obfuscate relevant details. For instance, as discussed with `m9g9` with respect to Figure 6, Pooled PC is quite precise, since it has access to all of the data, at the cost of significantly lower recall.
> >
> > On the other hand, we believe that the Structural Hamming Distance (SHD) is not an appropriate metric for our task since the MSS output is not necessarily a DAG but rather an equivalence class represented by a CPDAG with undirected edges; we are open to references for applicable SHD variants in this context, or if we misunderstood your suggestion.
> >
> >
> > [S1] Brouillard, P., Lachapelle, S., Lacoste, A., Lacoste-Julien, S., \& Drouin, A. (2020). Differentiable causal discovery from interventional data. Advances in Neural Information Processing Systems, 33, 21865-21877.
> >
> > [S2] Jaber, A., Kocaoglu, M., Shanmugam, K., \& Bareinboim, E. (2020). Causal discovery from soft interventions with unknown targets: Characterization and learning. Advances in neural information processing systems, 33, 9551-9561.
> >
> > [S3] K. D. Yang, A. Katcoff, and C. Uhler. Characterizing and learning equivalence classes of causal DAGs under interventions. Proceedings of the 35th International Conference on Machine Learning, 2018
> >
> > [S4] C. Heinze-Deml, J. Peters, and N. Meinshausen. Invariant causal prediction for nonlinear models. Journal of Causal Inference, 2018.

---

> > > ### Author Response · Authors · 2022-08-09
> > > **Rebuttal follow-up: are there remaining issues?**
> > >
> > > Dear `Rbq8`,
> > >
> > > Thank you again for reviewing our work and for the insightful feedback.
> > >
> > > We are writing to ask whether there are any remaining open issues that we may be able to respond to before the end of the discussion phase?
> > >
> > > We have addressed each of your initial four points and questions extensively and updated the revised manuscript accordingly. Specifically:
> > > 1. We now discuss [Brouillard et al. 2020] and [Jaber et al. 2020] in the context of I- and $\Psi-$ MECs in Secs. 3 & 5 (see l. 141 ff. & l. 233) and explore this connection theoretically in more detail in Appendix B.1. In short, under our assumptions, MSS indeed recovers a $\Psi$-MEC, but we respectfully disagree that Thm. 5.3 is straight-forward in that it provides a sufficient condition for recovering the unique true graph in terms of a meta-assumption on how environments arise (probabilistically sparse changes).
> > > 2. As we explain in our detailed response, the claim that many environments are needed to test for changing conditionals is incorrect. This is also not the main focus of our work, and MSS remains agnostic to which method is used to test for $P^1(X|Y)=P^2(X|Y)$.
> > > 3. We agree that causal sufficiency is a strong assumption, but an extension to non-Markovian systems is outside the scope of this work. While we were already transparent about this restriction in the first version (see l. 81), we have further emphasized this limitation in the discussion (see l. 388-389).
> > > 4. We have added a comparison in terms of F1 score to Figs. 5 and 6 in the revised manuscript: MSS clearly outperforms Pooled PC [Huang et al. 2020] and MC [Ghassami et al. 2017]. As explained in our response, neither [Brouillard et al. 2020] nor [Jaber et al. 2020] provide an implementation that handles our setting (unknown soft interventions, finite data).
> > >
> > > We think that this adequately addresses your initial review and further improved the paper. We were thus somewhat surprised that you acknowledged our rebuttal, but kept your original score of “Borderline reject” without further engaging with us. We would be grateful if you could kindly let us know what your remaining objections are.

---

> > > > ### Comment · Reviewer_Rbq8 · 2022-08-09
> > > > **Thank you for your response**
> > > >
> > > >  - Thanks for showing that the proposed method recovers $\Psi$-MEC. However, regarding Thm. 5.3, the authors mentioned in the rebuttal that ``There is a lack of study on when the equivalence class is solely the true DAG (unique structure identifiability)", I think it is not completely true as at least, in [1], there exist some identifiability results in Section 4.3. Although it is just for learning parents of a target variable, you can easily extend it to DAGs. The authors missed some main references in the first submission and it is needed to check the literature thoroughly for assessing the importance of Thm. 5.3 and also comparing with other similar results.
> > > >
> > > > - I think [Brouillard et al. 2020] support the setting in this paper but it is weird that their implementation does not work in this case. It is good to add an explanation in the revised version why it is not working with soft interventions.
> > > >
> > > > I read the rebuttal a few days ago but I did not have any meaningful questions from the authors. At that time, I decided to change my score after checking the changes in the revised version. Based on showing that the proposed method recovers $\Psi$-MEC, the result in the paper is now stronger and I decided to change my score from 4 to 5.
> > > >
> > > > [1] Peters, Jonas, Peter Bühlmann, and Nicolai Meinshausen. "Causal inference by using invariant prediction: identification and confidence intervals." Journal of the Royal Statistical Society: Series B (Statistical Methodology) 78.5 (2016): 947-1012.

---

> > > > > ### Author Response · Authors · 2022-08-09
> > > > > **Thank you for responding and increasing your score**
> > > > >
> > > > > Thank you for the response. We are glad to hear that "the result in the paper is now stronger" and that you have increased your score accordingly.
> > > > >
> > > > > Regarding the first point from your last comment: you wrote that
> > > > >
> > > > > > in [1], there exist some identifiability results in Section 4.3. Although it is just for learning parents of a target variable, you can easily extend it to DAGs.
> > > > >
> > > > > We are well aware of the identifiability results in [1] (ref. [42] in our manuscript, which was in fact one of the key motivations for our work), but strongly **disagree** with the last part of the statement **that these results are easily extended to learning DAGs**: as we write in l.135-139 (first paragraph of the Related Work Section 3), [1] relies on the assumption that the causal mechanism (structural equation) of a target variable $Y$ is unaffected by interventions and uses this to infer the causal parents of $Y$ from multiple, sufficiently heterogeneous environments. A naive extension of this approach for learning the full DAG would in turn treat each variable $X_i$ as the target $Y$ to infer its parents; the key obstacle to this, however, is that **this would entail the assumption that the mechanism of every single variable is unaffected, i.e., the i.i.d. setting**. *In this sense, our work could be seen as a meaningful extension of [1] for learning DAGs in which the assumption of perfect invariance is relaxed and replaced with a probabilistic version of sparsity of mechanism changes.*
> > > > >
> > > > > We tried to accurately and fairly place our work in the relevant literature, and remain grateful for the pointers to the two works we missed. If you have other relevant papers in mind, we would gladly hear about them.
> > > > >
> > > > > Regarding the second point: during the rebuttal and discussion phase, we have reached out to the main authors of [Brouillard et al. 2020] to discuss this. We have not yet asked for permission to release this private communication and have therefore not yet included it in the revised version yet, but will do so in the next revision as suggested. In short, we can confirm that their method and theory *does* support the setting of unknown imperfect soft interventions, but their implementation *does not*, mainly due to time & space constraints and computational challenges. An experimental comparison was therefore not feasible, but we will happily include a more polished version of the above in the next revision.
> > > > >
> > > > > [1] Peters, Jonas, Peter Bühlmann, and Nicolai Meinshausen. "Causal inference by using invariant prediction: identification and confidence intervals." Journal of the Royal Statistical Society: Series B (Statistical Methodology) 78.5 (2016): 947-1012.

---

### Author Response · Authors · 2022-08-02
**Summary response to all reviewers and the AC**

We sincerely thank all reviewers for their time and thoughtful feedback.

We are pleased to hear that
> "The idea of exploring sparse mechanism shifts in a non-parametric setting is original and of interest to the community" (`m9g9`)

and that our work
> "provides a sound method for using it by designing a score and analyzing it theoretically" (`qyjw`).

The reviewers state that the paper is "clear" (`m9g9`, `qyjw`) and "well-written" (`Rbq8`) and that the "theoretical results and assumptions have been rigorously stated" (`m9g9`), rating the presentation as "excellent" (4/4; `Rbq8`, `m9g9`) or "good" (3/4; `qyjw`) and the soundness as "good" (3/4; all three reviewers).

Both `m9g9` and `qyjw` also rate our contribution as "good" (3/4), while `Rbq8` considers it "fair" (2/4), citing a missing discussion of and comparison with two key prior works as the main critique.

We address this main concern and any other questions and comments in our detailed responses to each reviewer. Where relevant, we also include references to where these answers have been included in the revised manuscript.

Following the reviewers' suggestions, we also report the following suggested additional experimental and theoretical results:

- `Rbq8`, `qyjw`: In addition to precision and recall, we now also show F1 scores for our simulation results [Figures 5 and 6 in the main text].
- `Rbq8`: Formal characterization of the solution set as a $\Psi$-MEC [Line 233, Appendix Proposition B.1].
- `qyjw`: Test power for a multiplicative noise distribution in the bivariate setting [Appendix Figure 13].

(New or modified material is highlighted in yellow in the revised manuscript.)

We believe that the extended discussion of the placement of  our work in the related literature, triggered in particular by the comments of `Rbq8`, as well as the additional results and clarifications further improve the paper, and kindly ask the reviewers to take this into account when considering whether to adjust their score.

We will happily respond to any additional questions or comments by the reviewers during the discussion phase.

---

### Meta-Review · Area_Chair_zgCh · 2022-08-27

**Recommendation:** Accept
**Confidence:** Certain

**Metareview:**

The decision is to accept the paper.

The paper proposes a method for causal structure discovery that leverages an assumption about sparse mechanism shifts across multiple environments. The authors show that access to multi-environment data that satisfy this assumption can provide identification beyond standard equivalence classes. Based on reviewer / author discussions, the method seems novel, and is now well-contextualized within other literature in this area. The authors have thoroughly investigated near-alternatives, and while there are not direct comparisons in the paper, the authors cite good reasons for why this is the case. The paper is a solid contribution.

**Award:**

No

---

### Decision · Program_Chairs · 2022-09-14

Accept